# Association between various cathepsins and uterine leiomyoma: A Mendelian randomization analysis

Tingxiu Liu[1,2☯], Yuehan Ren[1,2☯], Junning Zhang[2,3☯], Hechun Yin[4], Zheng Zheng[1], Mingyue Zhang[1], Yan Liao[1,2], Liangliang Yang[1,2], Chang Liu[1,2], Xinmin Liu[1]*, Peiyu Yan[5,6]*

1 Department of Gynecology, Guang'anmen Hospital, China Academy of Chinese Medical Sciences, Beijing, China, 2 Beijing University of Chinese Medicine, Beijing, China, 3 Department of Oncology of Integrative Chinese and Western Medicine, China-Japan Friendship Hospital, Beijing, China, 4 Qi-Huang Chinese Medicine School, Beijing University of Chinese Medicine, Beijing, China, 5 Faculty of Chinese Medicine, Macau University of Science and Technology, Macao, Macao, SAR, China, 6 State Key Laboratory of Quality Research in Chinese Medicines, Macau University of Science and Technology Zhuhai MUST Science and Technology Research Institute, Macao, Macao, SAR, China

☯ These authors contributed equally to this work.
* XinminLIU9626@126.com (XL); teamzhang@126.com (PY)

**Data Availability Statement:** All relevant data are within the manuscript and its Supporting Information files.

## Abstract

Emerging evidence suggests a tentative association between cathepsins and uterine leiomyoma (UL). Previous investigations have predominantly focused on the role of cathepsins in the metastasis and colonization of gynecological malignancies. Still, observational studies may lead to confounding and biases. We employed a bidirectional Mendelian randomization (MR) analysis to elucidate the causative links between various cathepsins and UL. Instrumental variables (IVs) of cathepsins and UL within the European cohort were from extant genome-wide association study datasets. Sensitivity assessments was executed, and the heterogeneity of the findings was meticulously dissected to affirm the solidity of the outcomes. Our findings reveal the association between cathepsin B (CTSB) and an elevated risk of developing UL (all cancers excluded) [Inverse Variance Weighted (IVW) method]: OR = 1.06, 95%CI [1.02, 1.11], P = 0.008895711. Although the association does not persist after multiple testing or Steiger filtering, this finding adds to our understanding of the causal relationship between CTSB of various cathepsins and UL (all cancers excluded) and may herald new therapeutic avenues for individuals affected by this condition.

## Introduction

Uterine leiomyoma (UL), also known as uterine fibroid, is one of the most common types of benign gynecological tumors. ULs are usually formed by the proliferation of smooth muscle cells, fibroblast components, and fibrous extracellular matrix (ECM) [1]. Although patients may be asymptomatic, the presence of UL is often associated with symptoms such as irregular menstrual periods, abnormal uterine bleeding, pelvic pain, urinary problems, and

**Funding:** The author(s) received no specific funding for this work.

**Competing interests:** The authors have declared that no competing interests exist.

gastrointestinal symptoms. Nearly 70% of white women have ULs by the age of 50, basing on data from ultrasonic and pathology data [2]. However, these data alone may not be sufficiently accurate for UL [3]. The etiology of UL remains poorly elucidated. Risk factors for ULs include age, race, hormonal influences, reproductive history, genetic predisposition, lifestyle factors such as physical activity and diet, and exposure to endocrine disruptors and obesity [4]. Factors and molecular mechanisms that regulate UL development, growth, and regression remain undiscovered. Studies of family aggregation, prevalence, incidence, and racial differences have shown that genetic factors influence the risk of developing UL [5].

Emerging evidence suggests a tentative association between cathepsins and uterine leiomyoma [6]. As a suite of lysosomal proteases, cathepsins are pivotal in sustaining cellular equilibrium [7]. In the realm of gynecology, cathepsins assume a critical role. These enzymes are central to a plethora of cellular functions [8], ranging from metabolic processes and autophagy to signaling pathways and ECM turnover, underscoring their significance in health and disease. They orchestrate the degradation and remodeling of the ECM, marking them as critical targets for therapeutic intervention in ECM-associated pathologies such as multiple cancers [9].

Mendelian randomization (MR) is a genetic variant evaluation tool that explores the potential causal relationships between exposures and outcomes. Genetic variants serve as instrumental variables (IVs), randomly allocated at conception, thereby minimizing confounding factors to the greatest extent [10]. The design of MR studies is akin to randomized controlled trials due to the random assortment of alleles during gamete formation, providing a robust framework for assessing causality in genetic epidemiology. This method mitigates and reduces confounding and bias associated with observational studies, such as reverse causality [11].

## Materials and methods

### Study design description

This study accords with MR's three core assumptions [12]: IVs must correlate significantly with exposure and explain its variation, as confirmed by statistical tests such as the F-test. IVs are not related to other factors that influence both exposure and outcome, thus avoiding bias. The effect of the IV on the outcome is solely through the exposure, with no direct effect. This analysis utilizes data derived from multiple genome-wide association studies (GWAS) to unravel the bidirectional interrelations between ULs and various cathepsins. The forward MR strategy treats these cathepsins as predictors, assessing their influence on ULs. Conversely, the reverse MR methodology positions UL, as the predictor, scrutinizing its effect on various cathepsins. We clarify the construct of our bidirectional MR framework (Fig 1), designed to investigate the interplay between cathepsins and ULs, inclusive or exclusive of all cancer instances.

### Data sources and instrumental variables selection

Forward MR analysis was constructed by instruments for cathepsins, and instruments for UL were selected for reciprocal MR (https://gwas.mrcieu.ac.uk/, S1 Table). Similar to the MR analysis we completed earlier [14], appropriate UL IVs for MR analysis were selected from the latest centralized meta-analysis dataset, derived from cohorts of the FinnGen project (following the approved FinnGen study protocol Nr HUS/990/2017). This dataset encompasses 18,060 cases of individuals diagnosed with UL. Participants in the FinnGen study provided their written informed consent for biobank research, in alignment with the provisions of the Finnish Biobank Act [15]. All cathepsins' data came from an observational study that included 3301 individuals of nine different kinds of cathepsins from the European population, containing

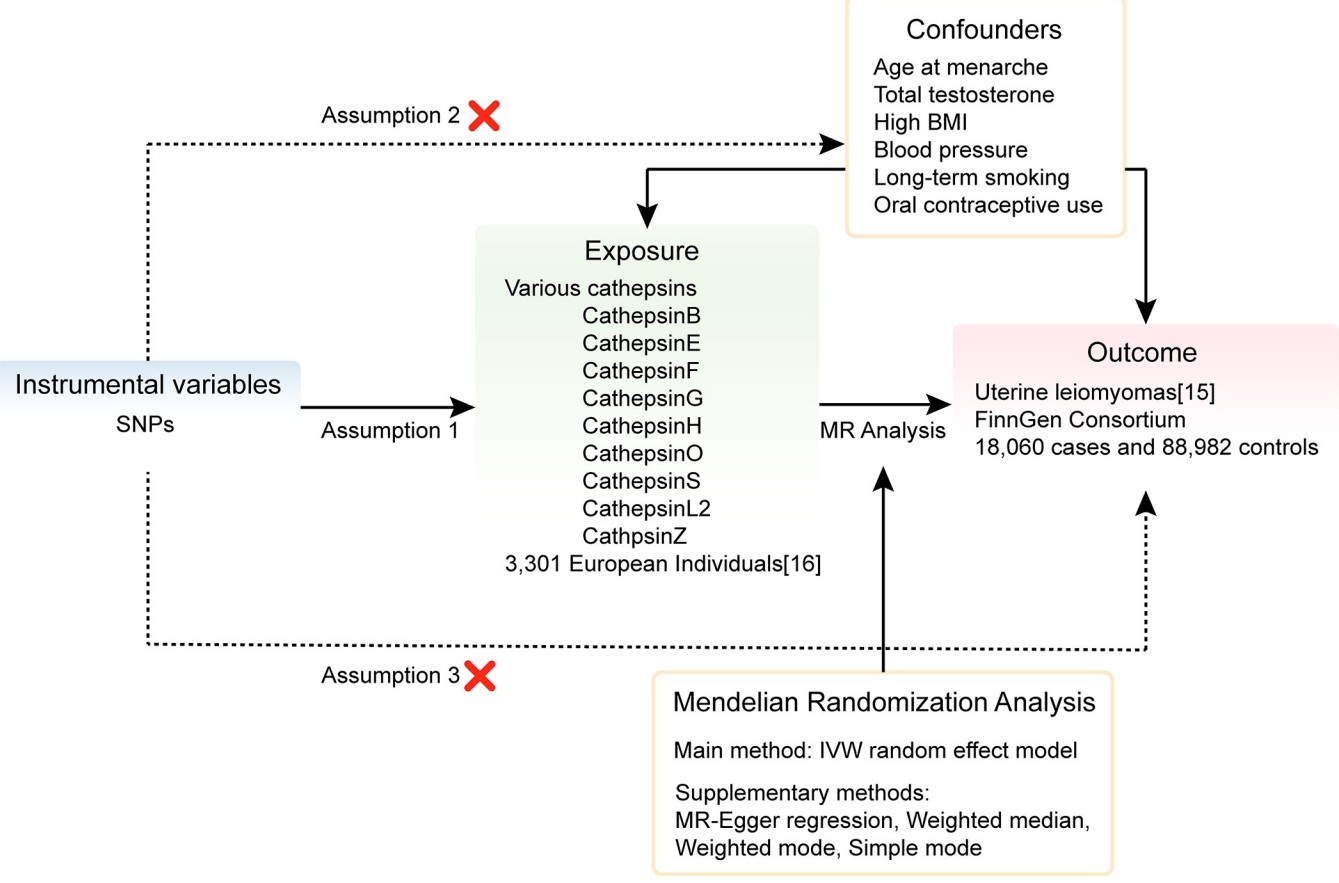

**Fig 1. The overview flowchart of hypothesis and study design.** A two-sample, bidirectional MR analysis using summary statistics from multiple genome-wide association studies (GWAS) to investigate the association between various cathepsins and uterine leiomyoma [13].

10,534,735 single nucleotide polymorphisms (SNPs) [16]. All study participants were of European ancestry. SNPs of cathepsins were selected with a threshold of P<5e×10–6 between cathepsin and IVs according to previous studies, and SNPs of UL were selected with a genome-wide significance threshold of P<5e×10–8 [17]. To account for the effect of linkage disequilibrium (LD) between SNPs, the LD clustering procedure was used (distance>10,000 kb, r2≤0.001) according to the European 1000 Genome Reference Panel. Then, those associating with the confounders of ULs were excluded manually according to the data from NHGRI-EBI catalogue (https://www.ebi.ac.uk/gwas/), assessing their association with any previously identified confounding traits ($P < 1e \times 10^{-5}$). When harmonizing data, we removed the palindromic SNPs with intermediate allele frequencies. We identified outliners by P<0.05 using MR-PRESSO global test (MR-PRESSO package, version 1. 0). The F-statistics were estimated to evaluate the instrument strength, and F-statistics < 10 indicates weak instrument strength [18]. The following methods were used for MR analysis: MR Egger, Weighted median, Inverse variance weighted (IVW), Simple mode and Weighted mode. IVW is the most sensitive method on what MR assumptions are violated [19]. To assess the positive results of CTSB on UL. The MR Steiger method was taken by testing the measurement error of the IVs [20]. As multiple MR analysis were performed, the result should be assessed with Bonferroni correction (P < 0.05/18). Sensitivity analysis was performed to evaluate the MR results and to ensure the validity of the research conclusions, including the MR-Egger intercept test

and the leave-one-out analysis, which tested for heterogeneity and horizontal pleiotropy. The presence of heterogeneity was assessed using Cochran's Q test. Heterogeneity analysis and Cochran's Q test were performed using the IVW method. Independent GWAS datasets should be used to confirm associations to ensure the reliability of the results. Statistical analysis was performed using the open-source TwoSampleMR package (version 0.5.7) in the R environment (version 4.2.3; R Development Core Team).

### Ethics statement

This study used publicly available de-identified data from participant studies that were approved by an ethical standards committee concerning human experimentation. No separate ethical approval was required in this study.

## Result

To evaluate the effect of different cathepsins on the risk of UL, we analyzed the overall risk of 9 cathepsins (B, E, F, G, H, O, S, L2, Z) and UL by two-sample MR. The results of the MR analysis suggested that high levels of cathepsin B (CTSB) increased the risk of UL (odds ratio (OR) = 1.05, 95% CI 1.00, 1.10, $P$ = 0.030). The other cathepsins did not increase the risk of UL (Fig 2).

### Analyzing sensitivity and defining the causal link between CTSB and UL

Sensitivity analysis of the bidirectional MR results of cathepsin and UL showed that all the results had no horizontal pleiotropy and heterogeneity. When the MR results of CTSB were analyzed by the retention method, it was found that the statistical difference disappeared after the removal of specific SNPs (OR = 0.99, 95% CI 0.93, 1.06, $P$ = 0.878), suggesting that CTSB had no causal effect on UL when CTSB was an exposure factor (Fig 2). The forest plot of the sensitivity analysis of CTSB on UL after the leave-one-out method can be found in S1 Fig, as well as the rsid of the excluded SNPs. The MR-Egger interception and MR-PRESSO global test of the above results are insignificant.

### MR subgroup analysis found association between cathepsin B on UL excluding all cancers

Further MR analysis was performed on the subgroup of UL (excluding all cancers) and cathepsin was selected as exposure. There was statistical significance of CTSB on the UL subgroup (OR = 1.06, 95% CI 1.01, 1.11, P = 0.009). However, the results of the remaining cathepsins showed no statistical significance. The inverse MR results showed no significant causal effect between UL (all cancers excluded) and CTSB (OR = 0.89, 95%CI [0.79, 1.00], P = 0.060) (Fig 3). In addition, the results showed a protective effect (OR = 0.93, 95% CI 0.88, 0.99, P = 0.019) of cathepsin O (CTSO) on ULs (all cancers excluded). Unfortunately, the protective effect is due to a single SNP (rs181844705) and cannot be considered statistically significant.

The bidirectional MR statistical results of all five methods for CTSB and UL (all cancers excluded) were performed (Fig 4). The CTSB on UL (all cancers excluded) scatter diagram (S2 Fig) and the leave-one-out plot of CTSB on UL (all cancers excluded) (S3 Fig) are provided in the supplementary materials.

## Discussion

Recent MR analysis have successfully unveiled the causal relationships between later menopause, reduced live births, decreased total testosterone levels, and increased risk of UL [21]. However, the current literature lacks MR analysis exploring the potential causal association

### Various Cathepsins on Uterine Leiomyoma

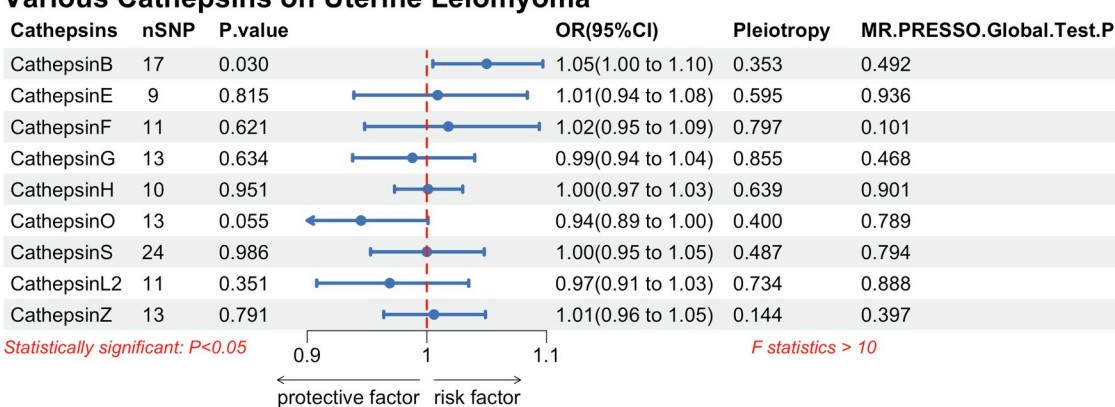

| Cathepsins | nSNP | P.value | | OR(95%CI) | Pleiotropy | MR.PRESSO.Global.Test.P |
|---|---|---|---|---|---|---|
| CathepsinB | 17 | 0.030 | | 1.05(1.00 to 1.10) | 0.353 | 0.492 |
| CathepsinE | 9 | 0.815 | | 1.01(0.94 to 1.08) | 0.595 | 0.936 |
| CathepsinF | 11 | 0.621 | | 1.02(0.95 to 1.09) | 0.797 | 0.101 |
| CathepsinG | 13 | 0.634 | | 0.99(0.94 to 1.04) | 0.855 | 0.468 |
| CathepsinH | 10 | 0.951 | | 1.00(0.97 to 1.03) | 0.639 | 0.901 |
| CathepsinO | 13 | 0.055 | | 0.94(0.89 to 1.00) | 0.400 | 0.789 |
| CathepsinS | 24 | 0.986 | | 1.00(0.95 to 1.05) | 0.487 | 0.794 |
| CathepsinL2 | 11 | 0.351 | | 0.97(0.91 to 1.03) | 0.734 | 0.888 |
| CathepsinZ | 13 | 0.791 | | 1.01(0.96 to 1.05) | 0.144 | 0.397 |

*Statistically significant: P<0.05*                          *F statistics > 10*

0.9      1      1.1

← protective factor | risk factor →

### Uterine Leiomyoma on Various Cathepsins

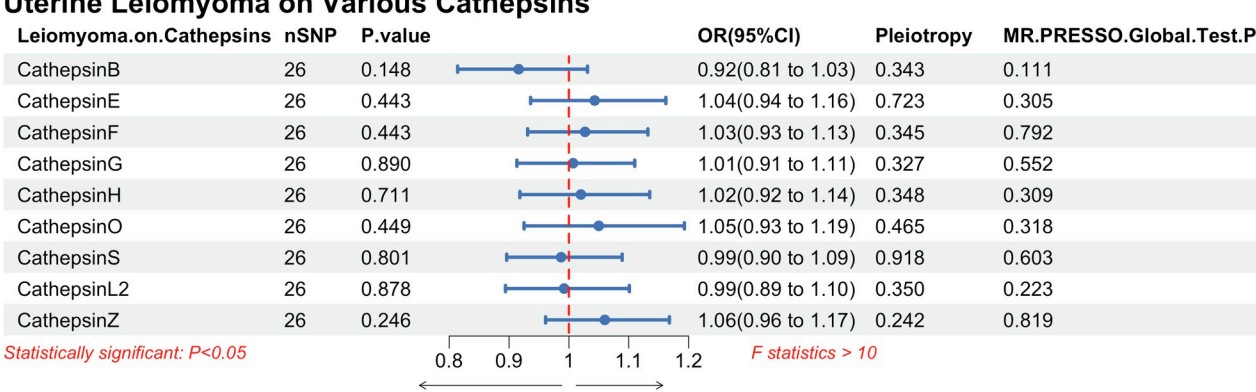

| Leiomyoma.on.Cathepsins | nSNP | P.value | | OR(95%CI) | Pleiotropy | MR.PRESSO.Global.Test.P |
|---|---|---|---|---|---|---|
| CathepsinB | 26 | 0.148 | | 0.92(0.81 to 1.03) | 0.343 | 0.111 |
| CathepsinE | 26 | 0.443 | | 1.04(0.94 to 1.16) | 0.723 | 0.305 |
| CathepsinF | 26 | 0.443 | | 1.03(0.93 to 1.13) | 0.345 | 0.792 |
| CathepsinG | 26 | 0.890 | | 1.01(0.91 to 1.11) | 0.327 | 0.552 |
| CathepsinH | 26 | 0.711 | | 1.02(0.92 to 1.14) | 0.348 | 0.309 |
| CathepsinO | 26 | 0.449 | | 1.05(0.93 to 1.19) | 0.465 | 0.318 |
| CathepsinS | 26 | 0.801 | | 0.99(0.90 to 1.09) | 0.918 | 0.603 |
| CathepsinL2 | 26 | 0.878 | | 0.99(0.89 to 1.10) | 0.350 | 0.223 |
| CathepsinZ | 26 | 0.246 | | 1.06(0.96 to 1.17) | 0.242 | 0.819 |

*Statistically significant: P<0.05*                          *F statistics > 10*

0.8   0.9   1   1.1   1.2

← protective factor | risk factor →

**Fig 2. Forest plot of multivariable bidirectional Mendelian randomization analysis [IVW method] for various cathepsins and uterine leiomyoma.**

between cathepsin and UL. This study is the first bidirectional MR study to investigate the association between nine cathepsins and UL in the European population. Our findings demonstrate a potential link between elevated levels of CTSB and the susceptibility to UL. Our analysis reveals no evidence of horizontal pleiotropy or heterogeneity in this association. While both cathepsin and UL GWAS data come from European cohorts, it is noteworthy that UL cohort data come from FinnGen, which is limited to Finland. Although the use of different database sources mitigates the potential bias arising from sample overlap, differences in the demographics of the cohorts must be duly taken into account. In addition, the sample size for the nine tissue proteases was relatively limited (3,301 cases), presenting a constraint.

## Pathogenesis of hysteromyoma: Degradation and remodeling of ECM and pathological angiogenesis

The aetiology and progression of UL are closely linked to the excessive accumulation of ECM and vascular abnormalities. As a benign tumor, UL is characterized by increased levels of

### Various Cathepsins on Uterine Leiomyoma (all cancers excluded)

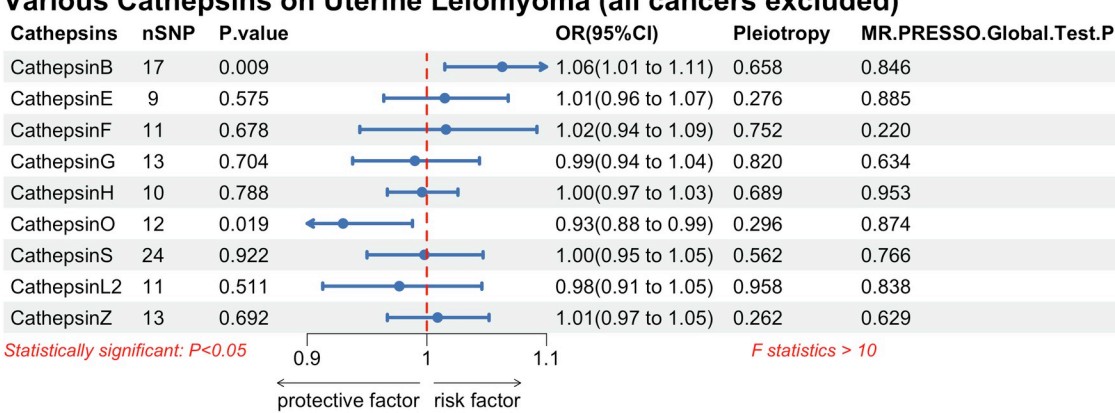

| Cathepsins | nSNP | P.value | OR(95%CI) | Pleiotropy | MR.PRESSO.Global.Test.P |
|---|---|---|---|---|---|
| CathepsinB | 17 | 0.009 | 1.06(1.01 to 1.11) | 0.658 | 0.846 |
| CathepsinE | 9 | 0.575 | 1.01(0.96 to 1.07) | 0.276 | 0.885 |
| CathepsinF | 11 | 0.678 | 1.02(0.94 to 1.09) | 0.752 | 0.220 |
| CathepsinG | 13 | 0.704 | 0.99(0.94 to 1.04) | 0.820 | 0.634 |
| CathepsinH | 10 | 0.788 | 1.00(0.97 to 1.03) | 0.689 | 0.953 |
| CathepsinO | 12 | 0.019 | 0.93(0.88 to 0.99) | 0.296 | 0.874 |
| CathepsinS | 24 | 0.922 | 1.00(0.95 to 1.05) | 0.562 | 0.766 |
| CathepsinL2 | 11 | 0.511 | 0.98(0.91 to 1.05) | 0.958 | 0.838 |
| CathepsinZ | 13 | 0.692 | 1.01(0.97 to 1.05) | 0.262 | 0.629 |

*Statistically significant: P<0.05*          *F statistics > 10*

protective factor ← → risk factor

### Uterine Leiomyoma (all cancers excluded) on Various Cathepsins

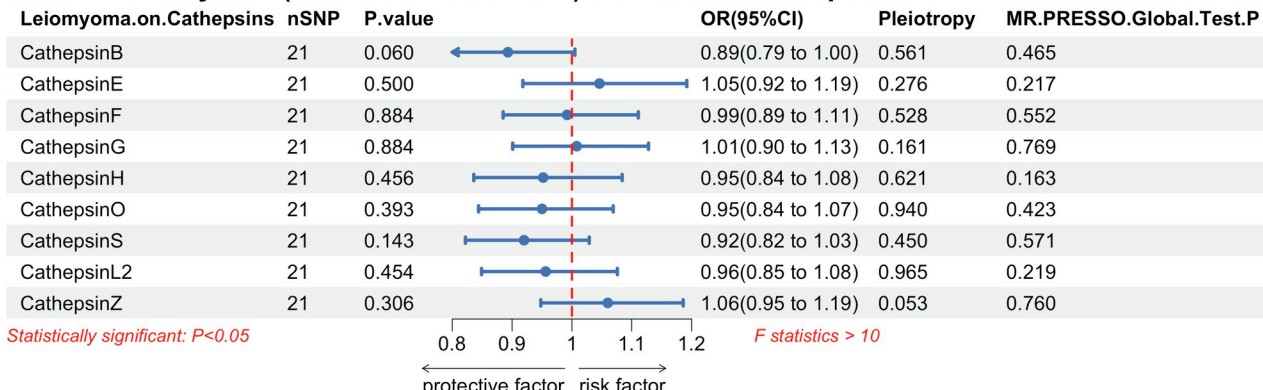

| Leiomyoma.on.Cathepsins | nSNP | P.value | OR(95%CI) | Pleiotropy | MR.PRESSO.Global.Test.P |
|---|---|---|---|---|---|
| CathepsinB | 21 | 0.060 | 0.89(0.79 to 1.00) | 0.561 | 0.465 |
| CathepsinE | 21 | 0.500 | 1.05(0.92 to 1.19) | 0.276 | 0.217 |
| CathepsinF | 21 | 0.884 | 0.99(0.89 to 1.11) | 0.528 | 0.552 |
| CathepsinG | 21 | 0.884 | 1.01(0.90 to 1.13) | 0.161 | 0.769 |
| CathepsinH | 21 | 0.456 | 0.95(0.84 to 1.08) | 0.621 | 0.163 |
| CathepsinO | 21 | 0.393 | 0.95(0.84 to 1.07) | 0.940 | 0.423 |
| CathepsinS | 21 | 0.143 | 0.92(0.82 to 1.03) | 0.450 | 0.571 |
| CathepsinL2 | 21 | 0.454 | 0.96(0.85 to 1.08) | 0.965 | 0.219 |
| CathepsinZ | 21 | 0.306 | 1.06(0.95 to 1.19) | 0.053 | 0.760 |

*Statistically significant: P<0.05*          *F statistics > 10*

protective factor ← → risk factor

**Fig 3. Forest plot of multivariable bidirectional Mendelian randomization analysis [IVW method] for various cathepsins and leiomyoma of uterus (all cancers excluded).**

collagen, fibronectin, laminin and proteoglycan. The blood supply for UL is mainly from the uterine artery. Several angiogenic factors, such as VEGF and IGF, play a role in the vascularization and growth of UL. Vascular growth factors can lead to the germination of capillaries and realize tumor revascularization. However, it was found that the microvessel density in the muscle layer around the UL increased significantly, forming dense vascular 'pseudocysts'. In contrast to the vascularization process of a malignant tumor, UL may show a decrease in the angiogenic response or an increase in factors that inhibit angiogenesis. At this point, the myometrium may compensate by growing blood vessels around the UL and increasing blood flow to the diseased area [22]. ECM can also regulate blood vessel homeostasis and maintain the typical morphology of capillaries [23]. Increased levels of ECM and myofibroblasts in ULs support the fibrotic characteristics of these tumors. Interestingly, ECM can be used as a repository of fibrogenic growth factors to enhance their activity by increasing their stability and prolonging their signaling duration [24].

## Cathepsin B on Uterine Leiomyoma (all cancers exluded)

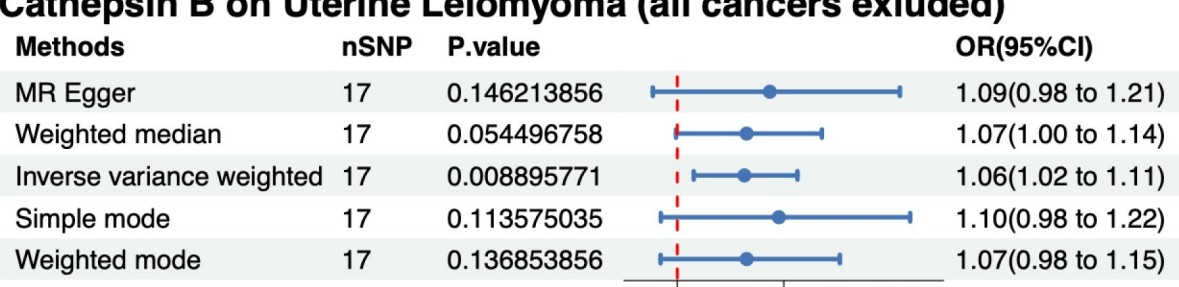

| Methods | nSNP | P.value | | OR(95%CI) |
|---|---|---|---|---|
| MR Egger | 17 | 0.146213856 | | 1.09(0.98 to 1.21) |
| Weighted median | 17 | 0.054496758 | | 1.07(1.00 to 1.14) |
| Inverse variance weighted | 17 | 0.008895771 | | 1.06(1.02 to 1.11) |
| Simple mode | 17 | 0.113575035 | | 1.10(0.98 to 1.22) |
| Weighted mode | 17 | 0.136853856 | | 1.07(0.98 to 1.15) |

*Statistically significant: P<0.05*     *F statistics > 10*

← protective factor   risk factor →

## Uterine Leiomyoma (all cancers exluded) on Cathepsin B

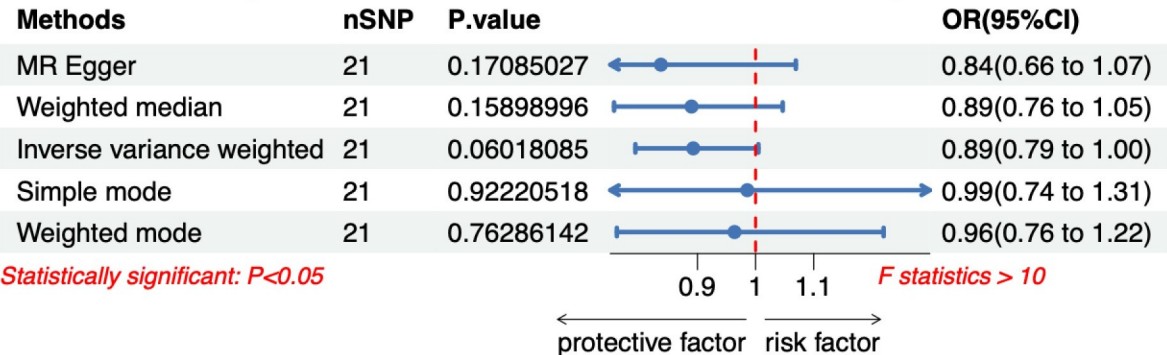

| Methods | nSNP | P.value | | OR(95%CI) |
|---|---|---|---|---|
| MR Egger | 21 | 0.17085027 | | 0.84(0.66 to 1.07) |
| Weighted median | 21 | 0.15898996 | | 0.89(0.76 to 1.05) |
| Inverse variance weighted | 21 | 0.06018085 | | 0.89(0.79 to 1.00) |
| Simple mode | 21 | 0.92220518 | | 0.99(0.74 to 1.31) |
| Weighted mode | 21 | 0.76286142 | | 0.96(0.76 to 1.22) |

*Statistically significant: P<0.05*     *F statistics > 10*

← protective factor   risk factor →

**Fig 4. Forest plot of multivariable bidirectional Mendelian randomization analysis for cathepsin B and uterine leiomyoma (all cancers excluded).**

### Possible role of cathepsins in the occurrence and development of UL

CTSB may influence the occurrence and progression of UL through ECM degradation, remodeling, and abnormal angiogenesis. The fundamental role of cathepsin involves the degradation and clearance of macromolecules, as well as foreign structures engulfed by digestive cells [25]. Emerging evidence underscores the pivotal role of cathepsin as a lysosomal enzyme in intracellular homeostasis and cellular responses. Lysosomes participate in immune response verification under cellular stress conditions [26]. CTSB, a lysosomal cysteine protease, possesses both endo- and exo-peptidase activities and is implicated in protein turnover [27]. CTSB acts via 3 isoforms: main transcript, main transcript lacking exon 2 or main transcript lacking exon 2 and 3.Its functions may encompass the regulation of angiogenesis, invasion, tumor proliferation, immune resistance, and cell differentiation [28]. Aberrant expression/ activity of CTSB is frequently associated with malignant tumor progression and metastasis [29]. The upregulation of pro-form CTSB secretion in tumors facilitates tumor progression and metastasis [30]. Studies have highlighted the overexpression of CTSB in the early stages of various cancer types [31–33]. Notably, the expression of CTSB in gastric and colorectal cancer correlates negatively with ECM, suggesting CTSB's involvement in ECM remodeling [34, 35]. CTSB is increasingly recognized as a potential therapeutic target in cancer treatment [36].

The evidence for the relationship between UL and cathepsin is limited. However, previous studies have reported higher levels of CTSB expression in tumors that invade the lamina propria than in submucosa [32], suggesting a potential link between UL and CTSB. The researchers suggest that the interaction between fibroblasts and type I collagen in breast cancer may enhance the translation or stability of CTSB [37]. In addition, the expression and secretion of CTSB are related to skeletal muscle differentiation [38]. Small et al. [39] observed that while the expression of CTSB increased, smooth muscle cells changed to a non-proliferative (contractile) state, which was consistent with the termination of vascular remodeling. In the process of VEGF-A-induced angiogenesis, with the transformation of venules into maternal vessels, the activity of CTSB increased significantly, which was consistent with the structural characteristics of the vascularization of UL [40]. The researchers suggest that increased cathepsin activity, especially CTSB, significantly promotes maternal angiogenesis in tumors [41]. In CTSB knockout mice, VEGF-A-induced angiogenesis and vascular permeability decreased significantly, which further emphasized the critical role of CTSB in this process.

## Research limitations

The IVs of the cathepsins did not pass the MR Steiger test. If multiple MR results were adjusted Bonferroni correction ($P < 0.0028$), then the CTSB's effect on ULs is insignificant ($P = 0.0089$). It could not be determined that CTSB is a direct risk factor for UL, and the effect of CTSB on UL still needs to be further investigated. To ensure the reliability of the results, we used independent GWAS datasets of UL. It was concluded that a bidirectional negative causal relationship between CTSB and ULs could be validated (S1 Table). Unfortunately, of the available cohorts, only the FinnGen database provided data on ULs (excluding all cancers), and therefore the positive conclusions we obtained could not be replicated.

## Conclusions

Our results support the effects of cathepsins on UL susceptibility and development by both extending and analyzing previous research on the processes underlying both conditions. By bidirectional MR analysis of different types of cathepsins and UL, we found that CTSB is a potential risk factor for UL, supporting previous research findings that the gene expression of CTSB is involved in the process of ECM remodeling, mediating the vascular microenvironment signal pathways. Limited causal relationship between UL and other cathepsins was found by bidirectional MR analysis. These findings have significant implications for patient counselling and raise the possibility of targeting CTSB as a therapeutic aspect in the management of UL.

## Supporting information

**S1 Table. The SNPs data of various cathepsins and UL.**
(DOCX)

**S1 Fig. Forest plot of the MR analysis of the causal relationship between various cathepsins on UL after leave-one-out method.**
(DOCX)

**S2 Fig. Scatter plot of the MR analysis of the causal relationship between CTSB on UL (all cancers excluded).**
(DOCX)

**S3 Fig. Leave-one-out plot of CTSB on UL (all cancers excluded).**
(DOCX)

## Acknowledgments

We appreciated all the genetics consortiums for making the GWAS summary data publicly available.

## Author Contributions

**Formal analysis:** Yuehan Ren, Junning Zhang.

**Writing – original draft:** Tingxiu Liu.

**Writing – review & editing:** Hechun Yin, Zheng Zheng, Mingyue Zhang, Yan Liao, Liangliang Yang, Chang Liu, Xinmin Liu, Peiyu Yan.

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
