## [Decision Letter · Decision Letter 0]

23 Jun 2024

PONE-D-24-11417Bidirectional mendelian randomization analyses explore the relationship between various cathepsins and uterine leiomyoma: Cathepsin B is the risk factor of uterine leiomyomaPLOS ONE

Dear Dr. Liu,

Thank you for submitting your manuscript to PLOS ONE. After careful consideration, we feel that it has merit but does not fully meet PLOS ONE’s publication criteria as it currently stands. Therefore, we invite you to submit a revised version of the manuscript that addresses the points raised during the review process.

the editorial office also wants to add the following:= albeit a large number of SNPs analysed, please ensure to use a relatively high genome wide significance threshold ie p < 10^-8 in order to produce an enough significant SNPs

= to include details on multiple testing correction to justify for an accurate data analysis

= it is necessary to use literature or a tool such as phenoscanner to validate whether the identified SNPs are associated with any other outcome or exposure

= interpretation and conclusions drawn must be supported by a robust methodology in order to fulfill our publication criteria 3 (https://journals.plos.org/plosone/s/criteria-for-publication)

We look forward to receiving your revised manuscript.

Kind regards,

David Chau

Academic Editor

PLOS ONE

Journal Requirements:

- https://doi.org/10.1017/thg.2023.48

(among others)

In your revision ensure you cite all your sources (including your own works), and quote or rephrase any duplicated text outside the methods section. Further consideration is dependent on these concerns being addressed.

5. Please amend the manuscript submission data (via Edit Submission) to include authors Dr. Tingxiu Liu, Dr. Yuehan Ren, Dr. Junning Zhang, Dr. Hechun Yin, Dr. Zheng Zheng, Dr. Mingyue Zhang, Dr. Yan Liao, Dr. Liangliang Yang, Dr. Chang Liu, and Dr. Peiyu Yan. 

Reviewers' comments:

Reviewer's Responses to Questions

**Comments to the Author**

1. Is the manuscript technically sound, and do the data support the conclusions?

Reviewer #1: No

Reviewer #2: Yes

2. Has the statistical analysis been performed appropriately and rigorously? 

Reviewer #1: No

Reviewer #2: Yes

3. Have the authors made all data underlying the findings in their manuscript fully available?

Reviewer #1: Yes

Reviewer #2: Yes

4. Is the manuscript presented in an intelligible fashion and written in standard English?

Reviewer #1: Yes

Reviewer #2: Yes

5. Review Comments to the Author

Reviewer #1: Summary:

Liu et al. present a well-motivated analysis on using Mendelian Randomisation (MR) to assess the genetically predicted causal effect of cathepsins on the risk of developing uterine leiomyoma UL). They used publicly available summary statistics to conduct two-sample MR with widely utilized software. They found that cathepsin B (CTSB) is associated with increased risk of uterine leiomyoma in sub-group analyses excluding cancer.

Cause for Major Concern:

Line 175: “Further MR analysis was performed on the subgroup of UL (excluding all cancers)” – please specify how this subgroup was identified. This is a major concern as I do not see a phenotype for UL (excluding all cancers) in the public FinnGen GWAS data release. I am only able to see GWAS results for three uterine cancers – carcinoma in situ of cervix uteri, malignant neoplasm of corpus uteri, and malignant neoplasm of cervix uteri, where the controls exclude all cancers. So please provide: 1. The FinnGen endpoint code (for example for UL it is: CD2_BENIGN_LEIOMYOMA_UTERI), 2. Number of cases and controls, and 3. An explanation from the FinnGen database of how this subgroup of UL (excluding all cancers) was selected. As this is the analysis with significant results, it is not possible to interpret the results presented without this information.

Other Major Comments:

1. The Introduction requires more references, for example: line 74, line 80, line 84, etc. that explain the prevalence, aetiology, and risk factors for UL.

2. Line 110: Please implement Steiger filtering, which is available in the TwoSampleMR package, to orient the causal relationship between cathepsins and UL. This can be in addition to the reciprocal MR results presented, but is necessary to ensure that instrument SNPs have stronger effects on the exposure than on the outcome. Ref: https://journals.plos.org/plosgenetics/article?id=10.1371/journal.pgen.1007081

3. Line 131: Please explain why such a high genome-wide significance threshold (P < 5E-6) was selected, when the standard in the field is P < 5E-8. Such a high threshold increases your chance of having spurious SNPs in the instrument, and also may lead to a weak instrument. Ref: https://www.nature.com/articles/s43586-021-00056-9

4. Line 132: Please explain the identification of independent SNPs in more detail. Did you use the LD-clumping procedure in the TwoSampleMR package? Please specify if so, and also move the distance for clumping (currently on line 131, which says 10,000 kb) to line 132.

5. Line 136: What are the “confounding traits” you tested? Please explain list which SNPs were removed due to association with confounders.

6. Line 138: Please briefly outline how MR-PRESSO identifies outliers, i.e. what criteria make a SNP an outlier?

7. Line 140: It is crucial to provide the F-statistics for all the instruments in all the various analyses.

8. Line 142: It is not correct that “IVW method is the gold standard of MR”, it depends on what assumptions are violated. If none are violated, then IVW is the most sensitive method. Please modify. Ref: https://www.ncbi.nlm.nih.gov/pmc/articles/PMC5506233/

9. Line 153: Since you are performing 9 MR tests (for all 9 cathepsins), you should adjust your P-value for multiple testing. Moreover, since you also perform reciprocal MR (another 9 tests), I would recommend Bonferroni correction (significance at P < 0.05/18) or for less stringency as the cathepsins are all somewhat correlated, FDR correction. If you apply Bonferroni, then none of the results will be significant.

10. Line 157: Please specify which SNPs were removed by sensitivity analysis, and which of the various sensitivity analyses you carried out (leave-one-out, heterogeneity tests, MR-PRESSO, etc.) caused this.

11. Lines 158-160: The leave-one-out analysis should produce 11 plots, as you have 11 SNPs in the instrument and each forest/scatter plot will display results from repeating the MR without one of the SNPs. You have only provided one forest/scatter plot – please check.

12. Lines 167-172: To my understanding, this is just re-stating the same things already reported in lines 158-161 (but the P-value reported is slightly different, 0.878 vs 0.876 – typo?) Remove this section or explain.

13. Line 178: “…remaining cathepsins showed no statistical significance.” Why is the protective association of Cathepsin O on UL excluding cancers (Figure 3) not discussed? The strength and significance of this association are comparable to CTSB (OR=0.93, P=0.019).

14. Figure 1:

a. Could the authors name some specific confounders they evaluated (as they mention in line 136)?

b. The “Exposure” box should cite the study from which genetic instruments were derived.

c. Minor comments: what does the symbol represent in the “Instrumental variables” box? In the “Mendelian randomization analysis” box, MR-PRESSO is mis-spelled.

15. Figures 2, 3, and 4:

a. Include instrument F-statistics in these plots

b. Adjust the P-value for multiple-testing correction

16. Figure 4: Please extend the axes to capture the full 95% CIs as these are not so large as to make the plots meaningless. In the top plot, include at least [0.95, 1.25] and in the bottom plot, [0.75, 1.30].

17. Supp. Table 1: It seems only one SNP for CTSB would pass a genome-wide significant threshold of P<5E-8, so I’m worried that the results are driven by a single SNP. The leave-one-out plots are also missing. You must also provide F-statistics so we can evaluate the strength of the instrument.

Minor Comments:

1. Please split the Introduction into three separate paragraphs – one on the aetiology of UL, one on cathepsins, and one on Mendelian Randomisation.

2. Line 107 & Figure 1 caption: Please correct that you use multiple GWASs as this is two-sample MR, not “a genome-wide association study (GWAS)” which means one single study.

3. Line 121: Please specify that you construct instruments for cathepsins for forward MR, and instruments for UL for reciprocal MR – it currently reads like you only did the latter.

4. Line 122: Which data release of FinnGen did you use?

Reviewer #2: The authors used bidirectional Mendelian Randomization to explore the causality of UL and found that Cathepsin B SNP is related to UL incidences. The authors used sound logic and methodology in this study. The data analysis was performed rigorously. However, I'd like to raise several questions for the authors:

1> In the abstract session, the authors stated that there are "Emerging evidence suggests a tentative association between cathepsins and uterine leiomyoma.", but could the authors give some citations to support this statement?

2>Can the authors specify the standard for eliminating the SNPs for the sensitivity analysis and quote studies using similar methodology?

3> Can the authors verify the findings using another independent cohort?

4> There's a grammar error in line 157-158.

6. PLOS authors have the option to publish the peer review history of their article (what does this mean?). If published, this will include your full peer review and any attached files.

Reviewer #1: No

Reviewer #2: No

---

## [Author Response · Author response to Decision Letter 0]

4 Aug 2024

According to the associate editor and reviewers' comments, we have made extensive modifications to our manuscript and Supplemented extra data to make our results convincing. Thank you for all your positive comments and valuable suggestions to improve the quality of our manuscript!

---

## [Decision Letter · Decision Letter 1]

16 Aug 2024

PONE-D-24-11417R1Association between various cathepsins and uterine leiomyoma: A Mendelian randomization analysisPLOS ONE

Dear Dr. Liu,

Thank you for submitting your manuscript to PLOS ONE. After careful consideration, we feel that it has merit but does not fully meet PLOS ONE’s publication criteria as it currently stands. Therefore, we invite you to submit a revised version of the manuscript that addresses the points raised during the review process.

We look forward to receiving your revised manuscript.

Kind regards,

David Chau

Academic Editor

PLOS ONE

Journal Requirements:

Reviewers' comments:

Reviewer's Responses to Questions

**Comments to the Author**

1. If the authors have adequately addressed your comments raised in a previous round of review and you feel that this manuscript is now acceptable for publication, you may indicate that here to bypass the “Comments to the Author” section, enter your conflict of interest statement in the “Confidential to Editor” section, and submit your "Accept" recommendation.

Reviewer #1: (No Response)

Reviewer #2: All comments have been addressed

2. Is the manuscript technically sound, and do the data support the conclusions?

Reviewer #1: Yes

Reviewer #2: Yes

3. Has the statistical analysis been performed appropriately and rigorously? 

Reviewer #1: Yes

Reviewer #2: Yes

4. Have the authors made all data underlying the findings in their manuscript fully available?

Reviewer #1: Yes

Reviewer #2: Yes

5. Is the manuscript presented in an intelligible fashion and written in standard English?

Reviewer #1: Yes

Reviewer #2: Yes

6. Review Comments to the Author

Reviewer #1: Summary:

Congratulations to Liu et al. for thoroughly addressing all my comments. My remaining concerns are mostly about the interpretation of results rather than the methodology.

Comments:

1. The association of cathepsin B with UL does not remain after:

a. Adjustment for multiple testing, or

b. Steiger filtering

Both of these give me cause for concern, as they indicate that this is likely a spurious association. While the authors mention this in the Limitations, these two major limitations should be presented in the Abstract, which currently has no indication that the result might not be real. Additionally, please make it clear that the presented finding is for cathepsin B on UL “excluding all cancers”.

2. The authors say they conducted independent validation of their findings using external datasets (in their response to reviewer #2). Why are these results not presented in the manuscript? I understand that they don’t have replication of the findings for UL-excluding cancer, but as the UL (not excluding cancer) result is also presented in the manuscript, it is worth showing the replication of this finding.

Reviewer #2: (No Response)

7. PLOS authors have the option to publish the peer review history of their article (what does this mean?). If published, this will include your full peer review and any attached files.

Reviewer #1: No

Reviewer #2: No

---

## [Author Response · Author response to Decision Letter 1]

20 Aug 2024

Journal Requirements:

>>Reply to Journal Requirements:

Thank you very much. We appreciate yout work and checked the cited papers, instead of replacing the current references, we added the relating informations (URLs) into the supplementary materials.

Reviewer #1：

Summary:

Congratulations to Liu et al. for thoroughly addressing all my comments. My remaining concerns are mostly about the interpretation of results rather than the methodology.

>>Reply:

Thank you very much for your valuable comments. We understand the concerns and have revised the manuscript to interpret our findings. The track changes of this revision are listed below.

Comments:

1. The association of cathepsin B with UL does not remain after:

a. Adjustment for multiple testing, or

b. Steiger filtering

Both of these give me cause for concern, as they indicate that this is likely a spurious association. While the authors mention this in the Limitations, these two major limitations should be presented in the Abstract, which currently has no indication that the result might not be real. Additionally, please make it clear that the presented finding is for cathepsin B on UL “excluding all cancers”.

>>Reply:

Thank you for your advice. Now in the revised manuscript, the concerns have been shown in both the Abstract and Limitations sessions. We have also clarified that the result presented is about cathepsin B on UL "excluding all cancers". 

Line 38:

Our findings reveal the association between cathepsin B (CTSB) and an elevated risk of developing UL (all cancers excluded) [Inverse Variance Weighted (IVW) method]: OR=1.06, 95%CI [1.02, 1.11], P=0.008895711. 

Line 40:

Although the association does not persist after multiple testing or Steiger filtering, this finding adds to our understanding of the causal relationship between CTSB of various cathepsins and UL (all cancers excluded) and may herald new therapeutic avenues for individuals affected by this condition. 

2. The authors say they conducted independent validation of their findings using external datasets (in their response to reviewer #2). Why are these results not presented in the manuscript? I understand that they don’t have replication of the findings for UL-excluding cancer, but as the UL (not excluding cancer) result is also presented in the manuscript, it is worth showing the replication of this finding.

>>Reply:

Thank you for your suggestions. Though we couldn’t replicate the result of UL-excluding cancer, the results of various cathepsins on an independent datasets of UL is worth to be presented. 

We described the missing information in both Method and Limitation session. The URLs of the independant datasets were replenished in the Supplementary materials (S1 Table).

Line 151:

Independent GWAS datasets should be used to confirm associations to ensure the reliability of the results.

Line 293:

To ensure the reliability of the results, we used independent GWAS datasets of UL. It was concluded that a bidirectional negative causal relationship between CTSB and ULs could be validated (S1 Table). Unfortunately, of the available cohorts, only the FinnGen database provided data on ULs (excluding all cancers), and therefore the positive conclusions we obtained could not be replicated.

According to the editor and reviewers' comments, we have made changes to our manuscript and Supporting informations. Thank you again for all your positive comments and valuable suggestions to improve the quality of our manuscript!

---

## [Decision Letter · Decision Letter 2]

29 Aug 2024

Association between various cathepsins and uterine leiomyoma: A Mendelian randomization analysis

PONE-D-24-11417R2

Dear Dr. Liu,

We’re pleased to inform you that your manuscript has been judged scientifically suitable for publication and will be formally accepted for publication once it meets all outstanding technical requirements.

Kind regards,

David Chau

Academic Editor

PLOS ONE

Additional Editor Comments (optional):

Reviewers' comments:

Reviewer's Responses to Questions

**Comments to the Author**

1. If the authors have adequately addressed your comments raised in a previous round of review and you feel that this manuscript is now acceptable for publication, you may indicate that here to bypass the “Comments to the Author” section, enter your conflict of interest statement in the “Confidential to Editor” section, and submit your "Accept" recommendation.

Reviewer #1: All comments have been addressed

2. Is the manuscript technically sound, and do the data support the conclusions?

Reviewer #1: (No Response)

3. Has the statistical analysis been performed appropriately and rigorously? 

Reviewer #1: (No Response)

4. Have the authors made all data underlying the findings in their manuscript fully available?

Reviewer #1: (No Response)

5. Is the manuscript presented in an intelligible fashion and written in standard English?

Reviewer #1: (No Response)

6. Review Comments to the Author

Reviewer #1: (No Response)

7. PLOS authors have the option to publish the peer review history of their article (what does this mean?). If published, this will include your full peer review and any attached files.

Reviewer #1: No

---

## [Editor Report · Acceptance letter]

3 Sep 2024

PONE-D-24-11417R2 

PLOS ONE

Dear Dr. Liu, 

I'm pleased to inform you that your manuscript has been deemed suitable for publication in PLOS ONE. Congratulations! Your manuscript is now being handed over to our production team.

Kind regards, 

on behalf of

Dr. David Chau 

Academic Editor

PLOS ONE